# Bending skyrmion strings under two-dimensional thermal gradients

Kejing Ran[1,2,8], Wancong Tan[1,8], Xinyu Sun[1,8], Yizhou Liu[3], Robert M. Dalgliesh [4], Nina-Juliane Steinke[5], Gerrit van der Laan [6], Sean Langridge [4], Thorsten Hesjedal [7] & Shilei Zhang [1] ✉

Magnetic skyrmions are topologically protected magnetization vortices that form three-dimensional strings in chiral magnets. With the manipulation of skyrmions being key to their application in devices, the focus has been on their dynamics within the vortex plane, while the dynamical control of skyrmion strings remained uncharted territory. Here, we report the effective bending of three-dimensional skyrmion strings in the chiral magnet MnSi in orthogonal thermal gradients using small angle neutron scattering. This dynamical behavior is achieved by exploiting the temperature-dependent skyrmion Hall effect, which is unexpected in the framework of skyrmion dynamics. We thus provide experimental evidence for the existence of magnon friction, which was recently proposed to be a key ingredient for capturing skyrmion dynamics, requiring a modification of Thiele's equation. Our work therefore suggests the existence of an extra degree of freedom for the manipulation of three-dimensional skyrmions.

Magnetic skyrmions are topological magnetization swirls, characterized by an integer winding number $Q$ which describes how often the real-space spin orientation wraps around a sphere in order-parameter space[1]. They interact with exotic particles[1-3], leading to emergent effects representing electromagnetism on the low-energy scale[4]. Their sensitive response to external angular momentum torque, stemming, e.g., from electric current[5,6] or a thermal gradient[5,7-11], makes them promising candidates for information storage[5,12-16], logic[17-19], and neuromorphic applications[20-22]. Therefore, the effective manipulation of skyrmions has been an active fundamental and applied research topic. On the other hand, while their effective manipulation is intriguing, the stability of the information has also to be guaranteed, which makes the study of possible detrimental effects equally important.

Recent studies uncovered a crucial aspect of the three-dimensional (3D) nature of skyrmions[23-31]. In general, for materials with finite thickness, skyrmions are described as vertically stacked 2D vortex planes, forming skyrmion strings in the third dimension[32,33]. From a manipulation point of view, the 3D nature of magnetic skyrmions hints towards an extra degree of freedom that requires attention. Instead of driving skyrmions to move within the lateral plane, the strings can also be bent or sheared if an appropriate effective force[34] can be identified. Though being elusive so far, such a manipulation mechanism would offer unique opportunities for bending, twisting, and cutting the skyrmion strings. Current-induced skyrmion string deformation has been studied theoretically[35,36]. For example, Yokouchi[35] reported a nonreciprocal Hall response which they ascribed to the asymmetric deformation of skyrmion strings due to the Dzyaloshinskii–Moriya interaction. On the other hand, Garst and collaborators have focused on the study of low-energy, nonlinear dynamics of skyrmion strings[37-39], exploring the effects of currents directed along the skyrmion string direction and finding that skyrmion strings are intrinsically unstable when a Zhang-Li spin transfer torque is applied along the string[39].

[1]School of Physical Science and Technology and ShanghaiTech Laboratory for Topological Physics, ShanghaiTech University, Shanghai, China. [2]College of Physics & Center of Quantum Materials and Devices, Chongqing University, Chongqing, China. [3]RIKEN Center for Emergent Matter Science (CEMS), Wako, Japan. [4]STFC, ISIS, Rutherford Appleton Laboratory, Didcot, UK. [5]Institut Laue-Langevin, Grenoble, France. [6]Diamond Light Source, Didcot, UK. [7]Department of Physics, Clarendon Laboratory, University of Oxford, Oxford, UK. [8]These authors contributed equally: Kejing Ran, Wancong Tan, Xinyu Sun. ✉e-mail: shilei.zhang@shanghaitech.edu.cn

These exotic skyrmion string manipulation operations are essential for creating new types of topological 3D objects. For example, the structural formation of chiral bobbers and emergent monopoles requires the skyrmion strings to be truncated[26,40,41], skyrmion braids require twists[28], hopfions require knots[42,43], and there are bifurcations that require splicing[44]. Moreover, it also offers an effective handle for manipulating information carriers in another dimension, meaning that the whole 3D skyrmion string can be regarded as a bit[45].

For manipulating skyrmion strings, we employ a thermal gradient as the driver[5,7–11,46], as illustrated in Fig. 1. The underlying principle of temperature gradient-induced 2D skyrmion motion can be understood in terms of magnon theory[8]. In the presence of thermal fluctuations, a random field $\mathbf{L}$ is introduced, whose correlation function is parameterized by the temperature $T$. Thus, the stochastic Landau-Lifshitz-Gilbert equation[47] reads: $\dot{\mathbf{m}} = -\gamma \mathbf{m} \times (\mathbf{H}_{\mathrm{eff}} + \mathbf{L}) + \alpha \mathbf{m} \times \dot{\mathbf{m}}$, where $\mathbf{m}$ is the normalized local magnetization, $\mathbf{H}_{\mathrm{eff}} = -\delta E/(M_S \delta \mathbf{m})$ is the local effective field that is a functional relating to the system's energy $E$, $M_S$ is the saturation magnetization, $\gamma$ is the gyromagnetic ratio, and $\alpha$ is the damping constant. Thiele's treatment[34], which assumes a steady-state motion of a skyrmion $\mathbf{m}(\mathbf{r}, t) = \mathbf{m}(\mathbf{r} - \mathbf{v}t)$, leads to the trace of the skyrmion described by its velocity $\mathbf{v}$. The equation of motion reads: $\mathbf{G} \times \mathbf{v} + \alpha \mathcal{D} \mathbf{v} = \mathbf{F}$, where $\mathbf{G} = -4\pi Q \hat{\mathbf{e}}_\perp / \gamma M_S^2$, with $\hat{\mathbf{e}}_\perp$ being the unit vector pointing normal to the 2D skyrmion plane; $\mathcal{D}$ is the dissipative tensor[48]; and $\mathbf{F}$ is the effective force acting on the skyrmion, whose exact form depends on the driver. At finite temperature $T$, the magnon dispersion broadens and the fluctuation field $\mathbf{L}$ enters into $\mathbf{F}$, leading to the stochastic force $\mathbf{F}^{\mathrm{sto}}$ that initiates Brownian skyrmion motion[22,49,50]. In a 1D temperature gradient $g_\parallel$, the magnon wavevector becomes unidirectional, resulting in a current $j_\parallel = (\pi/24)(k_B/\hbar s)^2 \bar{T} g_\parallel / \alpha$[8,51,52], where $\bar{T}$ is the spatially averaged temperature, and $s$ is the effective magnon velocity that can be regarded constant due to the linear dispersion relation for the skyrmion crystal (SkX) phase[52]. Such a magnon current

subsequently exerts a torque on the local moments, pushing the skyrmion towards the hot end[7,8,11,51]. Note that the direction of skyrmion travel is a materials property, and that skyrmions can also travel to the cold end as shown for Néel type skyrmions in heterostructures[10]. In the framework discussed here, $\mathbf{v}$ picks up a component $v_\parallel$ along $g_\parallel = \partial T/\partial z$, as well as a component $v_\perp$ due to the $\mathbf{G} \times \mathbf{v}$ term (see Fig. 1 for the definition of the coordinate system), called the skyrmion Hall effect[53,54]. The skyrmion Hall angle can be expressed as $\tan \Theta_{\mathrm{SH}} = v_\perp / v_\parallel \approx 2\alpha \mathcal{D}/G$[8].

A recent theoretical work by Weißenhofer et al. pointed out an extra mechanism that involves temperature as a key parameter in governing $\Theta_{\mathrm{SkH}}$[55]. Indeed, the fluctuation field $\mathbf{L}$ not only enhances the stochastic force, but also activates the thermal magnons as the heat bath into which the skyrmions sink. This in turn becomes a counter force for the skyrmion motion that is gauged by $\bar{T}$, termed magnon friction[55]. Under a 1D temperature gradient, the modified Thiele equation can be written as $\mathbf{G} \times \mathbf{v} + (\alpha \mathcal{D} + \eta \bar{T})\mathbf{v} = \mathbf{F}^{\mathrm{sto}} - \mathbf{F}^{\mathrm{magnon}}$, where $\mathbf{F}^{\mathrm{sto}}$ can be ignored for chiral magnets[5,8,9,11], and $\eta$ is a scalar parameter for the strength of the magnon friction. Dropping $\mathbf{F}^{\mathrm{sto}}$ and expanding $\mathbf{F}^{\mathrm{magnon}}$ leads to

$$\mathbf{G} \times \mathbf{v} + (\alpha \mathcal{D} + \eta \bar{T})\mathbf{v} = -Y\mathbf{G} \times \frac{\bar{T}}{\alpha} g_\parallel \hat{\mathbf{e}}_\parallel \ , \tag{1}$$

where $Y = \pi \gamma (k_B/\hbar s)^2/24$, and $\hat{\mathbf{e}}_\parallel$ is a unit vector pointing along the 1D thermal gradient. An immediate consequence of Eq. (1) is

$$\tan \Theta_{\mathrm{SkH}} = \frac{v_\perp}{v_\parallel} = \frac{\alpha \mathcal{D} + \eta \bar{T}}{bQ} \ , \tag{2}$$

where $b = -4\pi M_S/\gamma$. The key message from Eq. (2) is that $\Theta_{\mathrm{SkH}}$ acquires an additional dependence on the average temperature $\bar{T}$. Although being elusive so far, the experimental observation of such a pronounced temperature-dependent skyrmion Hall angle not only provides decisive

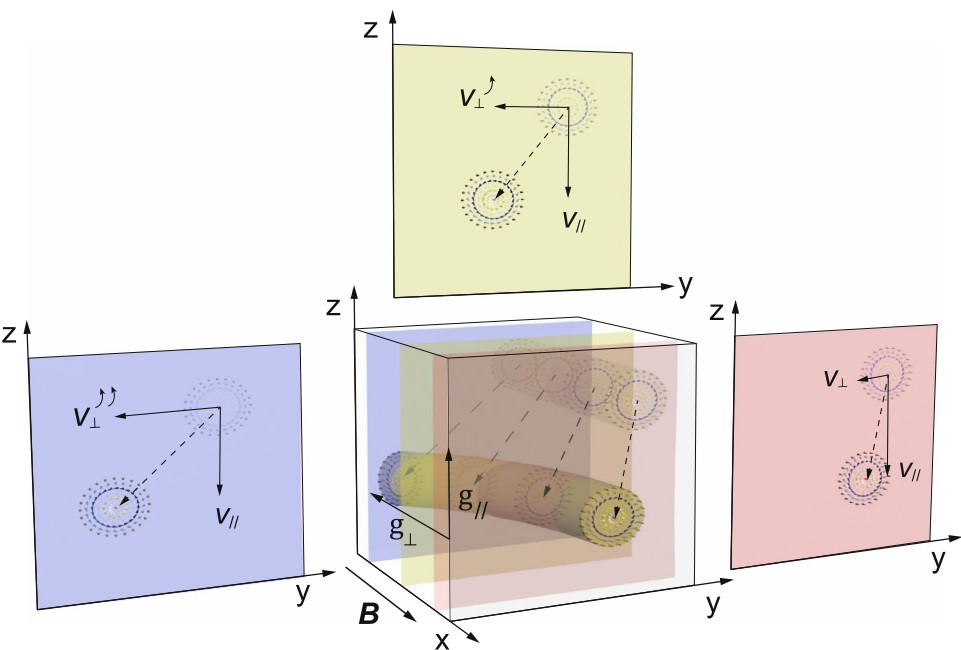

**Fig. 1 | Illustration of the skyrmion string bending mechanism in a 2D thermal gradient.** First, a 3D skyrmion string is created along the $x$ axis in a finite magnetic field $\mathbf{B} \| x$. Upon applying a primary temperature gradient $g_\parallel$ along the $z$ axis, the skyrmion acquires two orthogonal velocity components, i.e., $v_\parallel$ that is due to the thermal gradient drive, as well as $v_\perp$ that is due to the skyrmion Hall effect. It thus travels within the $yz$-plane, following a tilted trajectory, as shown in the three colored $yz$-slices. By switching on the secondary temperature gradient $g_\perp$ that is perpendicular to the $yz$-plane, different $yz$-slices experience a different average temperature $\bar{T}$, and therefore a different magnon friction. This results in a skyrmion Hall angle that is different for the different slices. Consequently, the skyrmion string experiences effective shear forces, resulting in its bending.

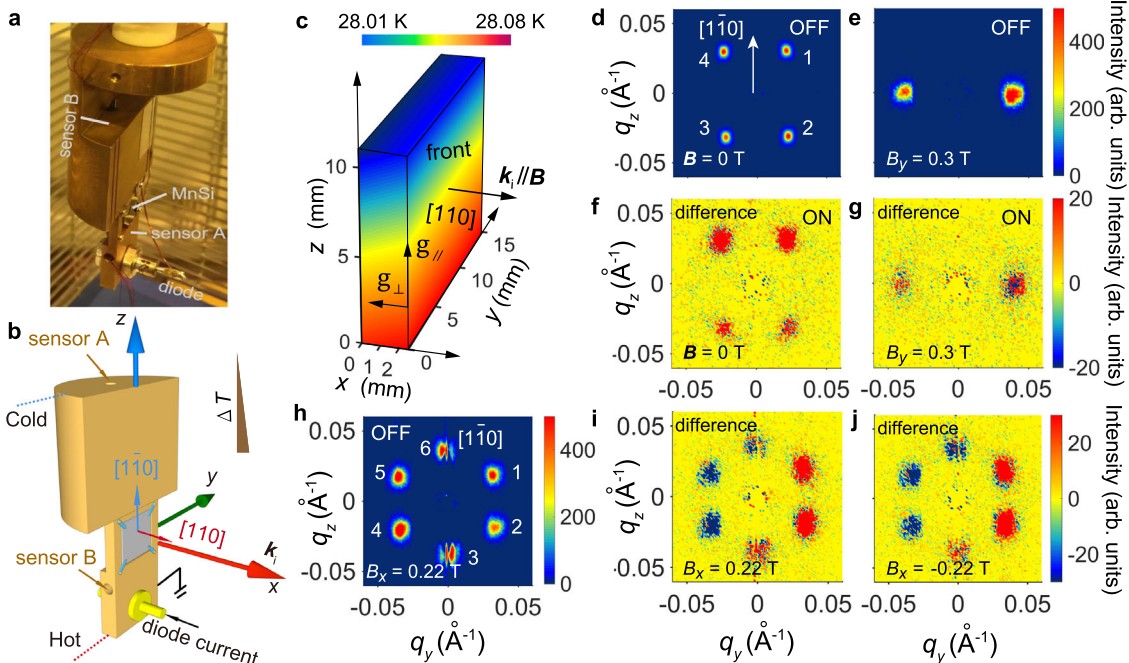

**Fig. 2 | 2D thermal gradient sample holder. a** Photograph and **b** design drawing of the SANS sample holder. **c** Experimentally calibrated temperature map inside of the MnSi bulk single crystal under the gradient drive. SANS patterns of the **d** helical (zero field) and **e** the conical phase (0.3 mT) with the temperature gradient turned off, and **f, g** the respective intensity differences in the SANS pattern after the temperature gradient has been switched on. **h** SANS pattern of the skyrmion lattice phase with its six characteristic peaks with the temperature gradient turned off, and **i, j** SANS difference after the temperature gradient has been switched on in an applied field of 0.22 and −0.22 mT, respectively.

evidence of the "hidden" magnon friction effect, but also offers an effective handle that can manipulate 3D skyrmion strings.

Figure 1 illustrates our strategy for exerting a force from the side on a 3D skyrmion structure, utilizing the magnon friction effect. First, applying a magnetic field along $x$ leads to a stack of $y$–$z$ skyrmion planes. In our tailored thermal configuration, a primary gradient $g_\parallel$ supplies the effective force that drives the skyrmions within each plane, traveling from the cold to the hot end with velocities $v_\parallel$ and $v_\perp$ and a finite $\Theta_{SkH}$. Meanwhile, a secondary temperature gradient $g_\perp$ is applied, as illustrated in Fig. 1. Due to the inhomogeneous $\bar{T}(x)$ profile, each slice picks up a gradually changing $\Theta_{SkH}$. This gives rise to an overall bending of the 3D skyrmion string, achieving the goal of manipulating skyrmions from the side.

## Results

Experimentally, a straightforward way of applying a 2D temperature gradient is shown in Fig. 2a, b, where a bulk single crystal MnSi with a thickness (along $x$) comparable to its lateral dimensions ($y$–$z$) is used. Next, a wedge-shaped temperature distribution within a sample, i.e., a 2D thermal gradient, is achievable by utilizing the intrinsic thermal conductivity of the material[56]. A thermal flow from the hot corner towards the cold corner (on the opposite side), forms a temperature gradient vector **g** that can be decomposed into two orthogonal directions: $g_\parallel = \partial T/\partial z$ as the main gradient, as well as $g_\perp = \partial T/\partial x$ as the secondary gradient perpendicular to it. As shown in Fig. 2c, by switching on **g**, we measured the temperature configuration inside of the MnSi crystal. It is found that each $y$–$z$-plane slice has the same $|g_\parallel| \approx 5.9$ mK/mm. However, the average temperature $\bar{T}$ linearly decreases by about 15 mK from the sample's front face to its back face (Fig. 2c).

Small angle neutron scattering (SANS) was performed on the LARMOR beamline at the ISIS neutron facility, using the time-of-flight technique. The incident neutron beam $\mathbf{k}_i$ is aligned along $x$, i.e., the [110] crystalline direction (see Fig. 2b, c). The magnetic phase diagram,

comprising of the helical (Fig. 2d), conical (Fig. 2e), and hexagonal SkX phases (Fig. 2h) are in good agreement with previous reports[57]. In order to obtain the SkX phase, a magnetic field **B** of 0.22 T was applied along $x$.

Figure 2h shows the diffraction pattern for the equilibrium SkX phase at 28.05 K with the gradient **g** turned off. Six skyrmion lattice peaks can be identified (labeled 1 − 6), with one pair locking along [1$\bar{1}$0]. When switching on the temperature gradient, we observe a change in the diffraction pattern. In order to remain in the skyrmion pocket, the maximum difference between the hottest and coldest end of the sample cannot exceed 0.5 K. The evolution of the SkX structure is captured by the difference of the diffraction patterns after and before applying the 2D thermal gradient. Figure 2i, j shows the difference in the $q_y$–$q_z$-plane after **g** was applied in a field of 0.22 and −0.22 mT, respectively, revealing a "half-red-half-blue" feature. In other words, peaks 1–3 become stronger, while peaks 4–6 become weaker, suggesting a SkX bending around the $z$ and $y$ axis, independent of the applied magnetic field direction. This bending mode is fundamentally different from the gradient-induced rotation in which the skyrmions are moving within the lateral plane[5,9,58]. Here, the skyrmion strings are "pushed" around from the side, therefore representing a new type of 3D dynamics. More surprisingly, the rotated SkX even remains after **g** is switched off, and the only way to "reset" the unrotated SkX phase is to reenter the skyrmion phase at $|g| = 0$ after warming up the system to above $T_C$.

In order to investigate the detailed response of the SkX to **g**, time-dependent measurements were performed. The time-integrated SANS count over 3600 s was decomposed into a series of frames, each being 15 s long. As shown in Fig. 3a, during this period, **g** was off for 900 s, after which **g** was ramped up to its maximum for 1800 s. Subsequently, the gradient was turned off, and the magnetic structure was monitored for another 900 s. We first applied such a protocol for the helical phase, in which four peaks (two helical domain pairs) lie within the $q_y$-$q_z$-plane along {111} (Fig. 3a, d). Before the gradient was turned on, the

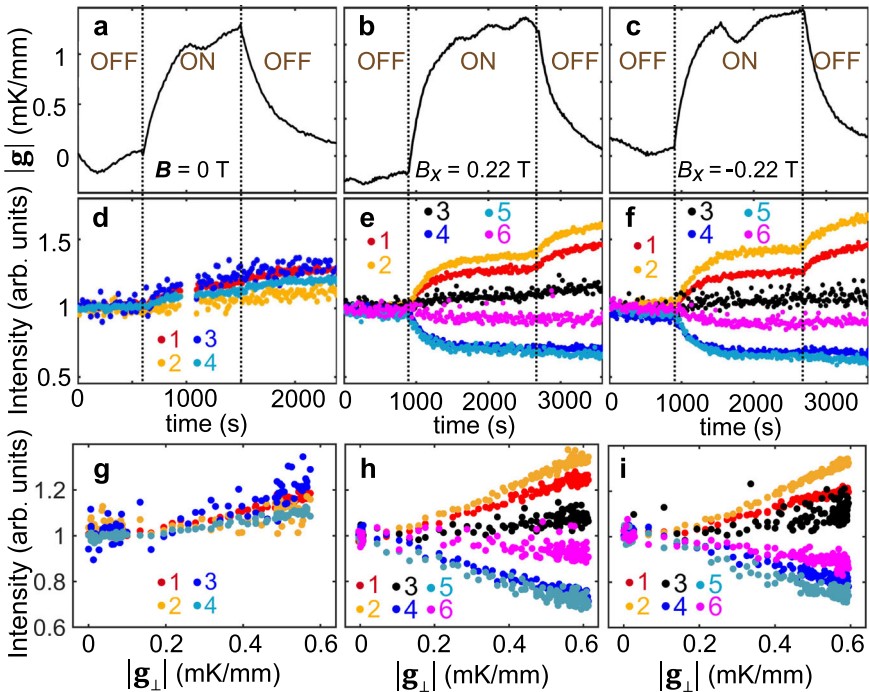

**Fig. 3 | Response of the helical and SkX phase to g through time-dependent measurements. a–c** Time-dependent applied temperature gradient amplitude |**g**|, and **d–f** corresponding peak intensities for the helical phase (zero field) and the SkX phase in a field of 0.22 and −0.22 mT, respectively. **g–i** The magnetic peak intensity as a function of $g_\perp$ for the states in (**d–f**), respectively.

peak intensities remained the same. The application of **g** increases the intensity of all four peaks due to the local evolution of the helical domains. For instance, due to the temperature inhomogeneity across the sample at finite **g**, the 1/3 domain pair may become more ordered, leading to an enhancement of their peak intensities. Therefore, it can be concluded that the 2D temperature gradient does not induce a bending in the helical phase, as shown in Fig. 3g under an alternative data representation (peaks' intensity versus $g_\perp$).

Next, we performed the same type of measurement for the SkX phase. The |**g**| profiles as a function of time (for $B_X$ = 0.22 and −0.22 mT, Fig. 3b, c, respectively) are very similar to the helical profile (Fig. 3a). Figure 3e, f shows the evolution of the six SkX peaks in response to |**g**|. For the first 900 s, the intensity of the six peaks remains unchanged and identical, representing a unrotated static SkX structure. Once **g** begins to ramp up, the bending starts. The change of the individual peak intensities track the amplitude of the gradient. Peaks 1/2 significantly enhance, while peaks 4/5 become weaker by the same amount. On the other hand, peaks 3/6 undergo smaller, yet noticeable changes (Fig. 3e, f). Figure 3h, i shows the direct relationship between the peaks' intensity and $g_\perp$, revealing the instantaneous response of the 3D skyrmion string structure to the transverse temperature gradient. Strikingly, when **g** ramps down during the last 900 s, the peaks maintain their 'inertia' by rotating further with the same sense of the bending. This counter-intuitive phenomenon is highly reproducible using the same measurement protocol. In other words, the 3D SkX will keep its bent structure without further external efforts to sustain it[59]. Note that this behavior is independent of the applied field direction, the behavior shown in Fig. 3e, f is almost indistinguishable.

By having access to most of the experimental parameters, it is possible to accurately extract the bending mode via SANS modeling. We first simulated the SANS contrast and then developed a data-fitting algorithm for retrieving the bending parameters based on the change in diffraction pattern. Note that the bending of the skyrmion lattice within the *y–z*-plane can be excluded as otherwise a rotation of the peaks in the $q_y$–$q_z$-plane would have been observed. Therefore, there are only two bending degrees of freedom to be considered, i.e., a

polar bending $\theta$ as well as an azimuthal bending $\psi$, as defined in Fig. 4k. Moreover, for **g** = 0 ($\theta$ = 0°, $\psi$ = 0°), the pristine SkX peak broadening in 3D reciprocal space was extracted, revealing a Gaussian distribution centered at $q_h = 0.035$ Å$^{-1}$. This is achieved by analyzing the wavelength-dispersed (elastic regime wavelength range 0.9–13.3 Å) diffraction data captured by the detector bank, from which the $q_x$-component is recovered, subsequently allowing for the reconstruction of the full 3D reciprocal space. The measured peak broadening profiles are due to the mosaicity of the imperfect long-range ordering of the SkX, which can be regarded as an intrinsic parameter of the SkX in our sample[60]. We then model a real-space SkX with $\theta$ = 0°, $\psi$ = 0°, from which the SANS reciprocal space configuration is simulated, taking into account the beamline parameters. The modeled peak broadenings are then normalized to the measured peak shape using a 3D Gaussian profile. The modified real-space SkX structure is then reconstructed via an inverse Fourier transformation, as shown in Fig. 4j. Next, the lattice is rotated about both the *y*- and *z* axis by $\theta$ and $\psi$, respectively, leading to a rotated magnetization configuration, from which a rotated SANS pattern is again calculated. Consequently, the simulated SANS difference pattern can be quantitatively compared with the experimental data. A least-square iteration then allows for the extraction of $\theta$ and $\psi$ at a particular time.

Figure 4a–i demonstrates the fitting process, showing how the bending angles were obtained at three particular time points. The first row of panels shows the measured SANS difference patterns at 885 s (**g** is off), 2685 s (**g** is on), and 3585 s (**g** is off again), respectively. The second row of panels shows the simulated SANS difference patterns, using the algorithm described above. Note that the patterns are optimized by iterating both $\theta$ and $\psi$, reaching a best-fit value. An excellent quantitative agreement between the data and the simulation is found, as shown by the comparison given in the third row of panels.

By performing SANS simulations, the exact bending angles $\theta$ (about the *y* axis) and $\psi$ (about the *z* axis) can be extracted, as shown in Fig. 4l. For **g** = 0, the SkX remains static, and when **g** is switched on, both polar and azimuthal bending start to occur (see Fig. 4j). The lattice bending scales with the amplitude of the gradient, and reaches

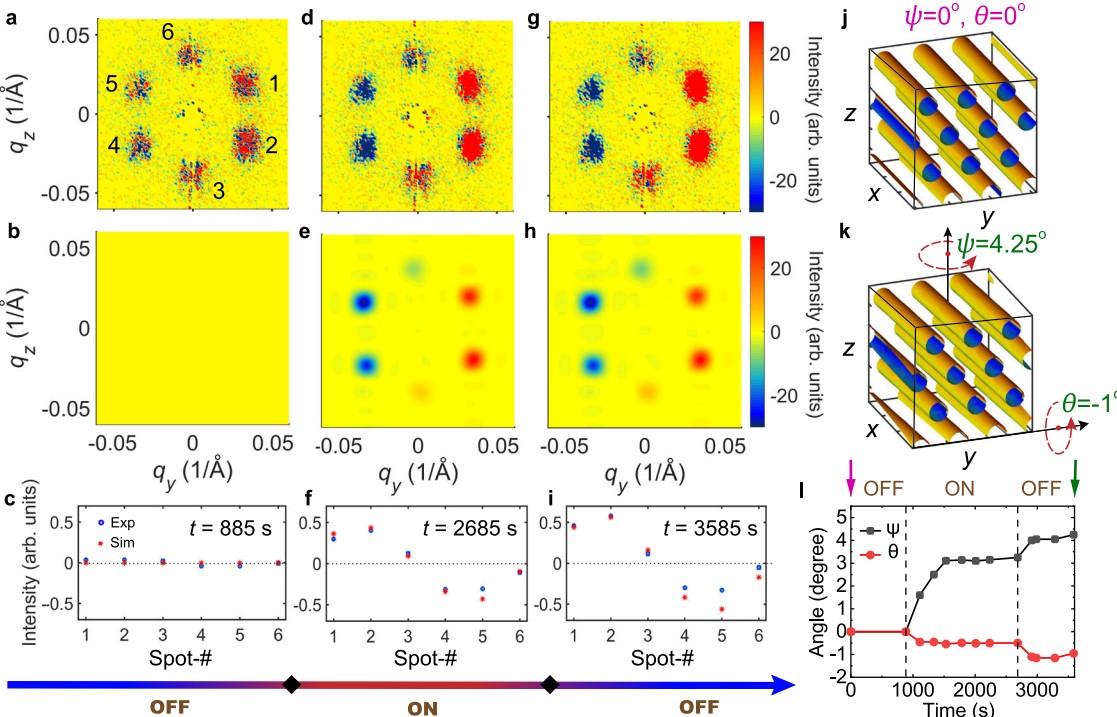

**Fig. 4 | Results of the SANS simulations, which allow for the extraction of $\theta(t)$ and $\psi(t)$.** **a**, **d**, **g** In the first row, the measured SANS difference patterns are shown. **b**, **e**, **h** In the second row, the simulated SANS difference patterns are shown, after the optimization of the bending angles. **c**, **f**, **i** The third row presents the comparison between the quantitative SANS difference intensity obtained in the experiment and in the simulation, suggesting the reliability of the fitting process. **j** Real-space SkX model in the absence of a thermal gradient. **k** Rotated SkX model (obtained from fitting the data) at $t = 3585$ s (equilibrium bending after removing the gradient). **l** Evolution of $\psi$ and $\theta$ as a function of time, covering the gradient off-on-off cycle.

values of $\psi \approx 3.4°$ and $\theta \approx -0.6°$, suggesting that the major bending is around $z$. After **g** ramps to its maximum value (at ~1500 s), the SkX maintains its bent state until the gradient is switched off at $t = 2700$ s. Interestingly, instead of recovering to its initial, unbent state, the SkX keeps bending as if the 2D temperature gradient was enhanced. This inertia-like behavior eventually ends and the bent state with $\psi \approx 4.25°$ and $\theta \approx -1°$ is the new equilibrium state (see Fig. 4k).

## Discussion

The experimental identification of the bending of the SkX in a 2D temperature gradient is interpreted by the following microscopic mechanism, sketched in Fig. 1. First, the 3D skyrmion string structure is treated as a stack of 2D skyrmion planes ($y$–$z$-planes). Within each 2D slice, $g_\parallel$ exerts a torque that drives the skyrmions towards the hot edge along $z$, leading to $v_\parallel$ and $v_\perp$. Equations (1) and (2) suggest that $v_\parallel$ is not a function of $\bar{T}$, but only proportional to $g_\parallel$. In our setup, $g_\parallel$ has the same value for all $y$–$z$-planes, leading to a uniform $v_\parallel$, i.e., all 2D skyrmions sheets moving toward -$z$ with the same velocity component. Such a moving mode does not cause a collective bending in $\theta$, which explains the observed minute bending value in Fig. 4l. The non-zero $\theta$ value for our experiment may be due to the $\bar{T}$-dependent Brownian motion in $v_\parallel$, caused by $\mathbf{F}^{sto}$, which is neglected in our model.

Turning to $v_\perp$, it is clear from Eq. (2) that at each $y$–$z$-plane, $v_\perp$ is strongly governed by $\bar{T}$. Consequently, the 3D skyrmion string assembly is determined by the $\bar{T}(x)$ configuration. Here, $\bar{T}(x)$ can be approximated by a linear profile, thus $v_\perp(x)$ can be regarded as a monotonic function. As shown in Fig. 1, at each $y$–$z$-plane, the 2D skyrmion sheet has a different, yet gradually decreasing skyrmion Hall deflection from back ($x = 0$) to front ($x = $ max). Overall, the skyrmion string is sheared, and thus the SkX exhibits a collective turning by $\psi$ about the $z$ axis. In such a model, it can be expected that under fixed $g_\parallel$ for all slices, $\psi = dv_\perp/dx =$ constant, i.e., the SkX undergoes a collective azimuthal bending. On the other hand, it can be derived that

$\tan \psi \propto \eta g_\perp$, which also agrees well with our experimental observations (see Supplemental Material S4). Nevertheless, the mechanism depicted in Fig. 1, together with the experimental results, provide straightforward evidence of the existence of magnon friction.

The temperature-governed skyrmion Hall angle can be well reproduced by stochastic micromagnetic simulations. For a 2D system with constant temperature gradient amplitude ($g_\parallel$ along $z$) yet varying temperature configuration, a skyrmion picks up both $v_\parallel$ and $v_\perp$ (Fig. 5a), exhibiting a finite $\Theta_{SkH}$. Most importantly, $\Theta_{SkH}$ has a strong $\bar{T}$ dependence, i.e., while the magnetic field is along $x$, the transverse skyrmion deflection decreases with increasing $\bar{T}$ (inset to Fig. 5a). Note that we chose a larger damping value and higher temperature in the calculations, in order to observe a pronounced skyrmion Hall effect. Thus, the simulated $\Theta_{SkH}$ is much larger than in the experiment. Nevertheless, the general $\Theta_{SkH}(\bar{T})$ relationship remains valid, regardless of the exact micromagnetic parameters. As **L** is numerically added to the calculation, the magnon friction was naturally taken into account[55], leading to a well-defined temperature-governed skyrmion Hall effect that is described by Eq. (1).

It is well-known that $\alpha$ and $\mathcal{D}$ are also functions of temperature and will therefore, in principle, modify the skyrmion dynamics[48]. However, they usually increase with temperature[48], which leads to simulation results [based on Eq. (1)] contradicting the experimental observations. Further, considering the very small value of the temperature gradient in our experiment of ~0.5 K, the effects of temperature on $\alpha$ and $\mathcal{D}$ were ignored.

Moreover, as shown in Fig. 5b, as long as the skyrmion motion was initialized by the temperature gradient, it will roughly maintain its 'inertia' even after the gradient is switched off. The off-gradient skyrmion trajectory seems to have a certain degree of randomness due to the vanishing magnon current, yet it still travels along its original direction. This is consistent with our experimental observations that the SkX keeps rotating once the gradient is switched off. Although the

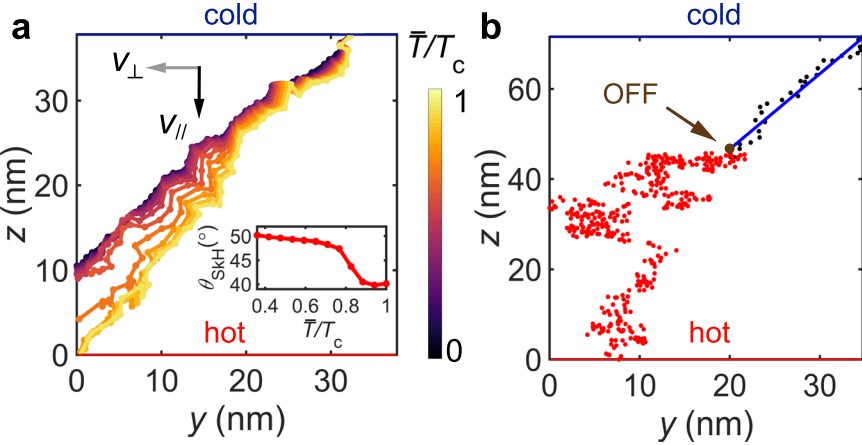

**Fig. 5 | Micromagnetic simulations of the 2D skyrmion trajectories. a** Simulation of the 2D skyrmion trajectories under the same $g_\parallel$ with varying $\bar{T}$. The inset shows the skyrmion Hall angle $\Theta_{\mathrm{SkH}}$ as a function of reduced temperature, $\bar{T}/T_{\mathrm{C}}$. **b** Simulation of the 2D skyrmion trajectory driven by $g_\parallel$ (blue) and after $g_\parallel$ is switched off (red). The plots in the $y$–$z$-plane are shown in reduced coordinates (divided by the skyrmion diameter, $d_{\mathrm{Sk}}$). The hot and cold ends of the sample (bottom and top, respectively) are indicated.

inertia effect can be well-captured by the stochastic simulation, the underlying mechanism requires further investigation. Qualitatively, the off-gradient dynamics is dominated by $\mathbf{F}^{\mathrm{sto}}$, and the inertia is related to the finite skyrmion effective mass[50,61,62]. Thus, the off-gradient skyrmion motion picks up randomness, a signature of Brownian motion. Moreover, for Brownian dynamics, both $v_\parallel$ and $v_\perp$ depend on $\bar{T}$, leading to both $\theta$ and $\psi$ bending. This explains why $\theta$ suddenly starts to increase after the gradient is switched off (Fig. 3c).

In summary, by setting up a 2D temperature gradient, we have identified a novel 3D SkX bending mode. The underlying mechanism is due to the temperature dependence of the skyrmion Hall angle, leading to the shearing of the skyrmion strings. This temperature dependence provides experimental evidence of the existence of magnon friction, a key factor that has been theoretically proposed, requiring a modified Thiele equation[55]. The confirmation of magnon friction has a profound impact on topological magnetism in general. First, the temperature-governed skyrmion Hall effect is a universal phenomenon, which applies to skyrmion dynamics resulting from all types of drivers. Second, the 3D SkX turning mechanism offers an effective handle for the manipulation of skyrmions in extended dimensions, which has been elusive so far. Moreover, our result highlights an issue for prospective applications as small temperature inhomogeneities are unavoidable in practical devices.

## Methods

### Experimental setup

Small angle neutron scattering (SANS) was performed on the LARMOR beamline at the ISIS neutron facility (Harwell, Oxfordshire, UK) using the time-of-flight technique. The incident neutron wavelength ranges from 0.9 to 13.3 Å. The cryostat is contained inside a superconducting vector magnet, which allows the magnetic field to be applied along any direction. The variable temperature insert (VTI, Oxford Instruments) is mounted inside the exchange-gas-based cryostat. Its rotational degree of freedom allows for performing rocking scans. In order to apply a wedge-shaped temperature gradient, a bespoke oxygen-free high thermal conductivity (OFHC) copper sample holder was made to accommodate our MnSi bulk sample. The MnSi single crystal was placed in a recessed part of the holder, and clamped down. Two Cernox sensors were mounted in order to monitor the real-time temperatures at the lower front end (heat source) and the upper back end (heat sink), respectively, along with the SANS patterns. Further details can be found in the Supplementary Section S1.

### Micromagnetic simulations

Micromagnetic simulations were performed using Mumax3[63]. A mesh size of $128 \times 128 \times 8$ and an elementary cell size of $1 \times 1 \times 1\,\mathrm{nm}^3$ were used. Open boundary conditions are applied in the $z$ direction, and periodic boundary conditions in $x$ and $y$. Further details can be found in the Supplementary Section S2.

## Data availability

The data that support the findings of this study are available from the corresponding author on request.

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

## Acknowledgements

The authors gratefully acknowledge the ISIS Neutron and Muon Source for beamtime under proposal RB1820289. S.Z. acknowledges funding from the National Key R&D Program of China (Grant Nos. 2022YFA1403600 and 2020YFA0309400), the Science and Technology Commission of the Shanghai Municipality (21JC1405100), the National Natural Science Foundation of China (Grant No. 12074257), and the Double First-Class Initiative Fund of ShanghaiTech University. K.R. acknowledges support from the National Natural Science Foundation of China (Grant No. 12374162). T.H. acknowledges support from the Engineering and Physical Science Research Council (UK) under grant EP/N032128/1.

## Author contributions

K.R., W.T., X.S., R.M.D., N.-J.S., G.L., S.L., T.H., and S.Z. performed the experiments and carried out the data analysis. W.T. and Y.L. carried out the supporting calculations. T.H. and S.L.Z. wrote the manuscript with input from all authors. All authors discussed the results and reviewed the manuscript.

## Competing interests

The authors declare no competing interests.
