## [Peer Review File · Nature Communications]

Reviewers' Comments:

Reviewer #1:

Remarks to the Author:

In the present paper, the authors consider the response of a 3d skyrmion lattice to a thermal gradient within the vortex plane and an additional thermal gradient along the skyrmion strings. The latter thermal gradient causes a variation of material properties along the strings which, in turn, results in different skyrmion Hall effects along the axis of the skyrmion string. As a result, the strings can be bent in a controlled manner.

The control of 3d magnetic textures is a challenging task and the interest in magnetic skyrmion is still unbroken. Here, the authors combine these two interesting fields and demonstrate in a technically challenging experiment a very simple yet beautiful idea how to manipulate the bending of 3d skyrmion strings. This is a remarkable result and another big leap on the way to control of 3d magnetic textures. I therefore recommend publication of this manuscript after minor corrections:

(1) The discussion of the literature is already good but there have been theoretical works already which discuss the dynamics of skyrmion strings. In particular, the bending has been discussed in <https://arxiv.org/abs/2005.04516>, but in fact not in great detail but merely as an observation. I therefore leave it to the authors whether or not they wish to include this reference. Moreover, several works on the dynamics of skyrmion lattices have been written by M. Garst and his collaborators, focusing on the magnon excitations in the vortex plane but also perpendicular to it. This group also published very recently more works on the effect of currents that are directed along the skyrmion string direction:

<https://journals.aps.org/prb/abstract/10.1103/PhysRevB.102.220408>

<https://journals.aps.org/prl/abstract/10.1103/PhysRevLett.131.066702>

These are important references as they find that skyrmion strings are intrinsically unstable if a Zhang-Li spin transfer torque is applied along the string. Since the thermal gradient force also takes the form of an effective Zhang Li torque, a similar intrinsic instability effect should be expected due to the thermal gradient along the skyrmion string. A discussion of this effect is detrimental for the interpretation of the measurements.

(2) The numerical simulations appear relatively sound. Strictly, the fluctuation dissipation theorem is violated and therefore the definition of the temperature model used here breaks down, but if one considers sufficiently fast local equilibration, this is still ok. Also, periodic boundary conditions in the x-y-plane are problematic as magnons propagate from the hot to the cold end also via the periodic boundary, which should better be suppressed by implementing absorbing boundaries with exponentially increasing Gilbert damping. This is not so dramatic, given that the simulation anyway does not produce new results but merely reproduces what has been shown previously in other works. What is missing, though, are simulations with the two orthogonal thermal gradients which prove that the interplay of the gradients causes a sizable effect as seen in the experiment and that this effect is not suppressed by the rigidity of the strings. I don't see any technical issues preventing simulations of this kind. There might be the problem that Mumax only supports a limited number of parameter regions, but coarse graining the regions could already help in combination with a much smaller Gilbert damping which would effectively blur the regions. Moreover, the limit for the number of regions is merely a hard-coded constraint in the source code which can be manually overwritten and elevated, there is no underlying technical reason why this limit was chosen, as far as I'm aware. I therefore ask the authors to please provide also these simulations which would be much more useful than the present simulation results for interpreting the results of the SANS measurements.

Reviewer #2:

Remarks to the Author:

Magnetic skyrmions are topological spin textures that have attracted intense attention from the magnetism community. To reveal emerging topological physics and device applications, the major challenge is to study the dynamics of which driven by different stimuli. This latter aspect has been frequently studied by different means, including current, current-induced spin torque, optical and thermal approaches. This manuscript, by using small angle neutron scattering, in the presence of 2 dimensional temperature gradient, authors find an intriguing being effect of skyrmion string. Through performing analytical calculation, the correlation of such an effect with the magnon friction is being established. I have found this manuscript interesting in general and worth of being publishing in Nature Communications, after addressing the following minor comments.

1: The direction of skyrmion motion driven by thermal gradients needs to be carefully considered. In the presence of temperature gradients, there are different sources of effective force that can act on the skyrmions, which could leads to motion of skyrmions either along or opposite to the direction of temperature gradients.

For a recent micromagnetic simulation work, please refer to Physical Review Applied 18, 024062 (2022).

2: I have noticed that the timescale of the dynamics of skyrmion bending effect, is in the minutes timescale. I am thus curious that are the controlling factors that determine such a big timescale?

3: I felt the experimental evidence of magnon friction effect is still lacking. I encourage authors perform more experiments to strengthen this part of discussion.

4: The temperature dependent skyrmion Hall angle requires a specific discussion. In the present experiment, the temperature gradient is extremely small, I am thus curious, how the 2 dimensional temperature to such a small degree could produce a pronounced effect. I suggest authors expand this part of discussion.

Reviewer #3:

Remarks to the Author:

In this manuscript, the authors discuss a bending of the skyrmion string under two dimensional thermal gradient. The observed bending behavior of the magnetic skyrmion diffractions is new and interesting. However, I consider the experimental setting has some problems as indicated below, and thus the physics discussion based on the detailed experimental setting information is uncertain.

First, the thermal gradient inside the MnSi sample, a key parameter in this experiment, is not experimentally determined. As an alternative way, the authors provide the simulation of the thermal gradient inside the MnSi sample as shown in Fig. 2c and discuss the detailed physics based on this simulation, but this is critical problem. We know that the simulation is useful tool if we prepare the ideal experimental setting. In short, in the case of this manuscript, if the authors discuss the thermal gradient inside the Cu holder, where two thermometers on cool and heat sides are attached, I agree the simulated thermal gradient is more reliable. However, in the current experimental setting, the authors discuss the thermal gradient inside MnSi sample attached by probably the mechanical clamping on the Cu sample holder (there is no detailed information about clamping in MS and Suppl.). In the case of the mechanical clamping, we need to consider the possibility that the thermal contact between the MnSi sample and Cu holder is not good by the thermally point contact problem. In that case, the thermal gradient inside the MnSi sample should be more complex than the ideal case. (If the authors use soldering or welding to connect the MnSi sample and Cu holder, the thermal contact is much better.)

Secondly, I consider the possibility of the bad thermal contact between the MnSi sample and Cu holder has been already shown in the experimental data. In Fig. 3e and 3f, the intensities of the SANS for 1 and 2 of diffractions increase after switching off the heater power. There is possibility that the origin of this increase is the thermal gradient between the Cu holder and MnSi sample after switching off the heater due to the bad thermal contact with them. The authors should

eliminate this possibility.

To answer the above criticism, I consider a best way is to measure experimentally the two dimensional thermal gradients inside MnSi sample. The next better way is that the authors should use not the simulation values but the information of the experimentally measured thermometers on the Cu holder in the manuscript, because the two thermometer values are experimentally reliable. The authors should not discuss the detailed physics based on the small thermal gradient inside the MnSi sample from the simulation result.

I consider if the authors correctly revise the above points, this manuscript is worthy to publish in Nature communications, because the physics discussion is so interesting. If the authors do not revise anything, I consider the current manuscript is not worthy to publish in any journal due to the lack of the experimental correctness.

Reply Letter to Referee's Comments

Kejing Ran,¹ Wancong Tan,¹ Xinyu Sun,¹ Yizhou Liu,² Robert M. Dalglish,³ Nina-Juliane Steinke,⁴ Gerrit van der Laan,⁵ Sean Langridge,³ Thorsten Hesjedal,⁶ and Shilei Zhang^{†1}

*¹School of Physical Science and Technology and
ShanghaiTech Laboratory for Topological Physics,
ShanghaiTech University, Shanghai 200031, China*

²RIKEN Center for Emergent Matter Science (CEMS), Wako, Japan

³STFC, ISIS, Rutherford Appleton Laboratory, Didcot OX11 0QX, United Kingdom

⁴Institut Laue-Langevin, 38042 Grenoble, France

⁵Diamond Light Source, Didcot OX11 0DE, United Kingdom

*⁶Department of Physics, Clarendon Laboratory,
University of Oxford, Oxford OX1 3PU, United Kingdom*

(Dated: March 22, 2024)

Abstract

We sincerely thank the three referees for their dedicated efforts in evaluating our manuscript. We are delighted that all three referees are recognising the novelty and significance of our work on the observation of effective bending of three-dimensional skyrmion strings in the chiral magnet MnSi in orthogonal thermal gradients. We value all the insightful comments and constructive suggestions provided by the referees, which have significantly contributed to the improvement of our work. In the point-by-point response below, we have addressed these recommendations and criticisms, and modified the manuscript accordingly. The referee comments are reproduced verbatim in blue, and our direct responses in black. We believe that we have been able to fully address all of the referees' concerns, and we are looking forward to the referees' recommendations for the publication of our manuscript in Nature Communications.

RESPONSE TO REFEREE 1

Comments:

In the present paper, the authors consider the response of a 3D skyrmion lattice to a thermal gradient within the vortex plane and an additional thermal gradient along the skyrmion strings. The latter thermal gradient causes a variation of material properties along the strings which, in turn, results in different skyrmion Hall effects along the axis of the skyrmion string. As a result, the strings can be bent in a controlled manner. The control of 3d magnetic textures is a challenging task and the interest in magnetic skyrmion is still unbroken. Here, the authors combine these two interesting fields and demonstrate in a technically challenging experiment a very simple yet beautiful idea how to manipulate the bending of 3d skyrmion strings. This is a remarkable result and another big leap on the way to control of 3d magnetic textures. I therefore recommend publication of this manuscript after minor corrections.

We sincerely thank the referee for the thoughtful and positive review of our manuscript. In particular, we thank for the referee's insightful comments about the possible instability of the skyrmion strings under a secondary temperature gradient. We have addressed the referee's comments with complementary experimental results, and performed systematic three-dimensional numerical simulations, as explained below. We would like to apologise for the delayed reply due to the additionally applied SANS beamtime, as such type of large-facility experiments requires long period in time.

(1) The discussion of the literature is already good but there have been theoretical works already which discuss the dynamics of skyrmion strings. In particular, the bending has been discussed in <https://arxiv.org/abs/2005.04516>, but in fact not in great detail but merely as an observation. I therefore leave it to the authors whether or not they wish to include this reference.

We thank for the referee for pointing out the earlier theoretical discussions about the inhomogeneity of the skyrmion strings. We have included this reference, which made our work complete.

(2) Moreover, several works on the dynamics of skyrmion lattices have been written by M. Garst and his collaborators, focusing on the magnon excitations in the vortex plane but also perpendic-

ular to it. This group also published very recently more works on the effect of currents that are directed along the skyrmion string direction:

<https://journals.aps.org/prb/abstract/10.1103/PhysRevB.102.220408>,

<https://journals.aps.org/prl/abstract/10.1103/PhysRevLett.131.066702>.

These are important references as they find that skyrmion strings are intrinsically unstable if a Zhang-Li spin transfer torque is applied along the string. Since the thermal gradient force also takes the form of an effective Zhang Li torque, a similar intrinsic instability effect should be expected due to the thermal gradient along the skyrmion string. A discussion of this effect is detrimental for the interpretation of the measurements.

We fully agree with the referee that the above mentioned two publications are important theoretical works about the instabilities of the strings under an effective Zhang-Li torque. Although the driving force is different in our case, it is reasonable to consider the possible shape instability in our experiment. After careful examinations, we can confirm that such instability effect can be neglected in our case.

First, as suggested in the above papers, if the skyrmion strings would have been ‘disturbed’ due to g_{\perp} , the skyrmion lattice would lose their crystalline order to certain degree. The mosaicity of the skyrmion lattice would lead to the decrement of the diffraction peak intensity for all six peaks. Nevertheless, we observed enhanced peak amplitudes for three of them instead (as shown in Fig. 3 in our main text). This suggests that the skyrmion lattice does not lose their order, but merely rotated.

Second, in order to study the possible effect of the string instability due to the temperature gradient, we performed additional beamtime in a complementary geometry. As shown in Fig.R1a, we used the same temperature gradient environment, i.e., g_{\parallel} is along z -direction. On the other hand, the magnetic field is along z for this time. In this case, the skyrmion lattice is formed within the xy -plane, while the primary temperature gradient g_{\parallel} is along the string. Such modified configuration amplifies the temperature gradient along the string. In other words, if the similar string instability would have happened, the skyrmion string lattice would have lost their crystalline order in a more pronounced manner, leading to obvious decrement of the magnetic peaks’ intensity. Nevertheless, under such configuration, we did not find noticeable change of the SANS pattern for the thermal gradient is ON nor OFF. The main conclusion is summarised in Fig. R1c and d. Therefore, it can be concluded that the primary gradient of g_{\parallel} does not have the similar Zhang-Li

spin-transfer torque effect that induces the strings instability.

Fig. R 1. (a) Illustration of the SANS geometry under our complementary measurement. We used the identical sample, the identical sample holder and the identical mounting orientation as that of in the previous submission, while the magnetic field is applied along z this time. (b) SANS pattern of the ‘in-plane’ skyrmion lattice phase, measured at 28.05 K, 0.22 T, without the temperature gradient. (c,d) The SANS pattern undergoes ignorable change over time before and after the temperature gradient is switched on.

This could be understood at a theoretical level that the temperature-gradient-induced effective current reads:

$$j_{\parallel} = (\pi/24)(k_B/\hbar s)^2 \bar{T} g_{\parallel} / \alpha . \quad (1)$$

(see line 60 in our main text). The key current source comes from the effective magnon velocity $s = d\omega/dk$, i.e., the slope from the magnon dispersion spectra $\omega(k)$. It is thus clear that s takes fundamentally different values along different orientations in momentum space, i.e., within the skyrmion lattice plane versus along the string, as the magnon dispersion spectra along these two directions varies a lot. On the other hand, the electric current j that gives rise to the spin transfer torque has the similar definition along the two directions. This could be the reason why the similar Zhang-Li effective torque applies differently in our case. While the detailed theoretical framework for the string instability due to the thermal gradient may worth further studies, we are confident that such effect does not take place in our experiments.

We have added a discussion of such aspect to main text on page 2. The extended paragraph now reads:

Current-induced skyrmion string deformation has been studied theoretically [35, 36]. For example, Yokouchi [35] reported a nonreciprocal Hall response which they ascribed to the asymmetric deformation of skyrmion strings due to the Dzyaloshinskii-Moriya interaction. On the other hand, Garst and collaborators have focused on the study of low-energy, nonlinear dynamics of skyrmion strings [37-39], exploring the effects of currents directed along the skyrmion string direction and finding that skyrmion strings are intrinsically unstable when a Zhang-Li spin-transfer torque is applied along the string [39].

(3) The numerical simulations appear relatively sound. Strictly, the fluctuation dissipation theorem is violated and therefore the definition of the temperature model used here breaks down, but if one considers sufficiently fast local equilibration, this is still ok. Also, periodic boundary conditions in the x-y-plane are problematic as magnons propagate from the hot to the cold end also via the periodic boundary, which should better be suppressed by implementing absorbing boundaries with exponentially increasing Gilbert damping. This is not so dramatic, given that the simulation anyway does not produce new results but merely reproduces what has been shown previously in other works. What is missing, though, are simulations with the two orthogonal thermal gradients which proof that the interplay of the gradients causes a sizable effect as seen in the experiment and that this effect is not suppressed by the rigidity of the strings. I dont see any technical issues preventing simulations of this kind. There might be the problem that Mumax only supports a limited number of parameter regions, but coarse graining the regions could already help in combination with a much smaller Gilbert damping which would effectively blur the regions. Moreover, the limit for the number of regions is merely a hard-coded constraint in the source code which can be manually overwritten and elevated, there is no underlying technical reason why this limit was chosen, as far as Im aware. I therefore ask the authors to please provide also these simulations which would be much more useful than the present simulation results for interpreting the results of the SANS measurements.

We appreciate the insightful comment from the referee. In regard with the boundary condition, we have compared the skyrmion dynamics between using periodic boundary condition and open boundary condition. We found that the skyrmion travelling trajectory undergoes minor difference

by setting up a relatively large mesh (e.g., 500 nm by 500 nm, while the skyrmion measures 18 nm), while considering the skyrmion moves in the region that is far away from the boundary.

Moreover, we fully agree that the simulations with two orthogonal thermal gradients were missing. We therefore established the codes based on Mumax3, and carried out these simulations. The new 3D simulation results under 2D orthogonal thermal gradient have now been added to the revised supplementary information, and are pasted in below for convenience. Indeed, as shown in Fig. R2, the micromagnetic simulations show that the interplay of the gradients causes a sizeable effect on the skyrmion string bending, as it is particularly evident in panel (b) below.

Fig. R 2. (a) Micromagnetic simulation result of the 3D skyrmion string bending effect in a 2D thermal gradient, showing the skyrmion's string structures at two time stamps, i.e., $t = 0$ before the orthogonal temperature gradient is switched on; and $t = 30$ ns after g is switched on. (b) Side-view of (a), projecting the same information at xy -plane. The rotation of the string around z -axis (Ψ) due to g_{\perp} can be clearly observed. (c,d) Skyrmion trajectories in the front and back yz -planes respectively.

RESPONSE TO REFEREE 2

Comments:

Magnetic skyrmions are topological spin textures that have attracted intense attention from the magnetism community. To reveal emerging topological physics and device applications, the major challenge is to study the dynamics of which driven by different stimuli. This latter aspect has been frequently studied by different means, including current, current-induced spin torque, optical and thermal approaches. This manuscript, by using small angle neutron scattering, in the presence of 2 dimensional temperature gradient, authors find an intriguing being effect of skyrmion string. Through performing analytical calculation, the correlation of such an effect with the magnon friction is being established. I have found this manuscript interesting in general and worth of being publishing in Nature Communications, after addressing the following minor comments.

We thank the Reviewer for the support for our manuscript, and appreciate the referee's valuable comments and suggestions. In particular, the referee encouraged us to put more efforts on the discussions about the magnon friction. We have reproduced the data by performing an additional SANS beamtime. In this way, we focused more on the quantitative data analysis based on the magnon friction model, which further improved the quality of our work. Please find our point-to-point reply in the following. Moreover, we would like to apologize for the delayed reply due to the additionally applied SANS beamtime, as such type of large-facility experiments requires long period in time.

(1) The direction of skyrmion motion driven by thermal gradients needs to be carefully considered. In the presence of temperature gradients, there are different sources of effective force that can act on the skyrmions, which could leads to motion of skyrmions either along or opposite to the direction of temperature gradients. For a recent micromagnetic simulation work, please refer to *Physical Review Applied* 18, 024062 (2022).

We thank the referee for the concern about the skyrmions moving direction. We fully agree with the referee that under 1D temperature gradient, a skyrmion may either move towards the cold or the hot end, depending on the practical materials properties. There also could be different sources of the skyrmion motion under a temperature gradient such as the magnon torque and entropy torque.

For example, as the referee points out, in *Physical Review Applied* 18, 024062 (2022), as well as in *Nat. Electron.* 3, 672 (2020), the skyrmions are stabilized by the surface/interfacial DMI from the heterostructure thin film systems. It is established both in theory and experiments that the skyrmions move towards the cold end.

On the other hand, in chiral magnets where the skyrmions are formed in crystalline system with bulk-type DMI, it is commonly recognised that they are driven to the hot end instead. This was theoretically treated in *Phys. Rev. B* 86, 054432 (2012), *Phys. Rev. Lett.* 111, 067203 (2013) based on the chiral magnets. Such model was supported by experimental observations in chiral magnets, e.g., in *Science* 330, 1648 (2010) for MnSi, as well as in *Nat. Commun.* 12, 5079 (2021) for Cu₂OSeO₃. Therefore, we are confident that in our MnSi single crystal, the skyrmions move towards the hot end under the major temperature gradient of g_{\parallel} . In the revised manuscript, we have included these literatures.

(2) I have noticed that the timescale of the dynamics of skyrmion bending effect, is in the minutes timescale. I am thus curious that are the controlling factors that determine such a big timescale?

We thank the Reviewer for highlighting this point, and we apologise for the unconventional data representation, which has caused the misunderstanding. In fact, in our experiment, the skyrmion bending process occurs rather instantaneously. In our main manuscript, we plotted both the temperature gradient and the magnetic peaks' intensity as a function of the time sequence, which records the history of the diode current application. As both the temperature gradient and the SANS pattern follow instantaneously with the diode current, one is able to obtain the quantitative relationship between g_{\perp} and the magnetic peaks' intensity. As shown in Fig. R3, we have therefore replotted Figure 3 in the main text to show the intensity (normalised) over the gradient and not over time. We wish the new data representation clarifies the skyrmion string's bending process.

(3) I felt the experimental evidence of magnon friction effect is still lacking. I encourage authors perform more experiments to strengthen this part of discussion.

We thank for the referee's insightful suggestions, and we fully agree with the referee that the physical picture can be made much clearer by performing in-depth experiments and analysis. We have applied and performed additional beamtime, from which the quantitative analysis of the

Fig. R 3. (a-c) Time-dependent applied temperature gradient amplitude $|g|$, and (d-f) corresponding peak intensities for the helical phase (zero field) and the SkX phase in a field of 0.22 and -0.22 mT, respectively. (g-i) The magnetic peak intensity as a function of g_{\perp} for the states in (d-f) respectively.

magnon friction has been performed as the following way:

As shown in Fig. R4(a), we consider two slices of the yz -planes that are separated by a distance of dx . Note that we employed the identical geometry as that of in the main text. Therefore, the skyrmion Hall angle $\Theta_{\text{SkH}} = v_{\perp}/v_{\parallel}$ is x -dependent if the magnon friction is taken into account. By switching on the 2D thermal gradient g_{\parallel} and g_{\perp} , the skyrmions start to move. Let us assume that our analysis is performed after a reasonable time constant. In this case, the skyrmions in both two slices must have travelled equal distance of s_{\parallel} along z -direction, but different s_{\perp} component long y due to the temperature-dependent skyrmion Hall angle. As sketched in Fig. R4(a), the skyrmion Hall angle for the two slices reads [see Eqn. (2) in our main text]:

$$\tan \Theta_{\text{SkH}}(x) = \frac{s_{\perp}(x)}{s_{\parallel}} = \frac{\alpha \mathcal{D} + \eta \bar{T}(x)}{bQ}, \quad (2)$$

$$\tan \Theta_{\text{SkH}}(x + dx) = \frac{s_{\perp}(x + dx)}{s_{\parallel}} = \frac{\alpha \mathcal{D} + \eta \bar{T}(x + dx)}{bQ}, \quad (3)$$

where $b = -4\pi M_S/\gamma$. Subtracting the two equations leads to

$$\frac{dl}{s_{\parallel}} = \frac{\eta g_{\perp} dx}{bQ}, \quad (4)$$

where $dl = s_{\perp}(x + dx) - s_{\perp}(x) = dx \tan \Psi$, as sketched in Fig. R4(a), and it is clear that $\bar{T}(x + dx) - \bar{T}(x) = g_{\perp} dx$. We therefore obtain the key relation between the observed string rotation angle Ψ and the applied gradient of g_{\perp} :

$$\tan \Psi = \frac{\eta g_{\perp} s_{\parallel}}{bQ} \propto \eta g_{\perp}. \quad (5)$$

Fig. R 4. (a) The geometry of the skyrmion dynamics at two adjacent lateral planes, separated by dx . The sketch shows the skyrmion string's bending angle Ψ to the geometrical properties of the system. (b) Experimentally measured $\tan \Psi$ as a function of g_{\perp} .

Equation (5) essentially establishes the magnon friction model, and can be interpreted as the following: Without the magnon friction bath, one would expect that $\tan \Psi = 0$ at all g_{\perp} values. If the magnon friction effect would take place, the strings' bending angle of $\tan \Psi$ is a linear function of the applied g_{\perp} , whereas the slope measures the friction's amplitude η . Figure R4(b) shows the measured $\tan \Psi$ value as a function of g_{\perp} . It is thus clear that such a linear profile unambiguously validates the existence of the magnon friction effect. We have attached the above discussions in our revised manuscript, as well as the updated supplementary material.

(4) The temperature dependent skyrmion Hall angle requires a specific discussion. In the present experiment, the temperature gradient is extremely small, I am thus curious, how the 2 dimensional

temperature to such a small degree could produce a pronounced effect. I suggest authors expand this part of discussion.

We agree with the Reviewer that the applied temperature gradient is very small. However, our choice of the gradient amplitude is restricted by the extremely small temperature window (~ 1.5 K) for the skyrmion phase pocket for MnSi (as well as most of the bulk chiral magnets). In other words, if the temperature difference between any two regions is larger than 1.5 K, the skyrmion phase would have been vanished. In fact, we observed a change in θ of $\sim 1^\circ$ and Ψ of less than 5° , which are discernible, but also not so large in value.

Nevertheless, as suggested by Eqn. (5), the observed bending angle Ψ is in fact not correlated with the skyrmion Hall angle Θ_{skH} at all, whereas it is merely governed by the geometrical properties. For example, let us assumed that the skyrmion has traversed the entire length of the crystal, i.e., $s_{\parallel} = L_z$. In this case, for larger crystal size with larger L_z value, the lattice rotation Ψ becomes more pronounced. Therefore, the experimentally observable bending angle can be ascribed to such geometrical argument.

RESPONSE TO REFEREE 3

Comments:

In this manuscript, the authors discuss a bending of the skyrmion string under two dimensional thermal gradient. The observed bending behavior of the magnetic skyrmion diffractions is new and interesting. However, I consider the experimental setting has some problems as indicated below, and thus the physics discussion based on the detailed experimental setting information is uncertain. To answer the above criticism, I consider a best way is to measure experimentally the two dimensional thermal gradients inside MnSi sample. The next better way is that the authors should use not the simulation values but the information of the experimentally measured thermometers on the Cu holder in the manuscript, because the two thermometer values are experimentally reliable. The authors should not discuss the detailed physics based on the small thermal gradient inside the MnSi sample from the simulation result. I consider if the authors correctly revise the above points, this manuscript is worthy to publish in Nature communications, because the physics discussion is so interesting. If the authors do not revise anything, I consider the current manuscript is not worthy to publish in any journal due to the lack of the experimental correctness.

We thank the reviewer for the critical comments and valuable advices. We fully agree that the characterisation of the local temperature configuration inside of MnSi would have been ideal, however, at the time of the submission of the paper, we were not able to perform such task due to the limited beamtime. We did explore optical and scanning probe microscopy-based thermometry, but both approaches were either unsuitable for our sample configuration or did not have high enough resolution. As the Reviewer will see in our response below, in an additionally applied beamtime, we managed to reproduce the same experiment using the identical sample and the identical temperature-gradient holder. This time, by using a Pt electrode-based thermometer, we are able to provide the missing experimental data, which eventually makes our claim sound.

We first introduce our method that obtains the experimentally measured temperature gradient inside of the MnSi crystal, after which we answer the referee's questions. The measurement of the local temperature employed the sensitive response of the Pt resistivity to the temperature. For example, as shown in Fig. R5(b), the patterned microstructures of the Pt electrodes has a well-defined $\rho(T)$ profile. In particular, the slope of the resistivity curve is reasonably large within the temperature range between 27 K and 29 K. This offers an ideal calibration table for one to obtain

Fig. R 5. (a) The configuration of the sample mounting mechanism, as well as the configuration of the Pt electrodes. (b) Typical Pt resistivity curve at the temperature range of interest. (c) The $\rho(T)$ curves for the four channels that are patterned on the four corners of the as-measured MnSi crystal. (d) Experimentally calibrated temperature map inside of the MnSi crystal while the temperature gradient is on.

the exact temperature value based on the resistivity reading.

In our case, we first polish the surface of the MnSi bulk crystal, on top of which an insulating SiO₂ layer with 5 nm thickness was deposited using magnetron sputtering. Subsequently, Pt electrodes (4 nm in thickness) were fabricated using thin film deposition and lithography techniques. We have fabricated four groups of the Pt electrodes (labeled as channel 1 to 4 as shown in Fig. R5), located at the four corners of the crystal, i.e., top (channel 1) and bottom (channel 2) on the front surface, as well as the top (channel 3) and bottom (channel 4) on the back surface, respectively.

Next, we calibrate the $\rho(T)$ curves for all four channels in-situ without applying the diode current, i.e., the temperature distribution across the MnSi crystal is homogenous. Figure R5(c) shows the four $\rho(T)$ profiles measured simultaneously. Note that the four curves do not necessarily overlap due to the shape difference of the Pt electrodes, which is inevitable during the fabrication process. We would like to emphasize that despite of the fact that the four $\rho(T)$ curves exhibits minor difference, it does not affect the precision of the temperature reading. This is because each individual channel serves as its own calibration table.

At $\bar{T} = 28.05$ K, upon switching on the diode current, we immediately observed the resistivity change for all four channels. By converting the readout ρ -values into T -values, the temperature reading at the four corners are shown in Fig. R5(d). As our MnSi sample is a good single crystal, it is reasonable to assume a continuous thermal distribution throughout the entire bulk sample.

We therefore obtain the experimentally measured temperature distribution in Fig. R5(d). It is thus clear that both g_{\parallel} and g_{\perp} well exist, validating the fact that we indeed applied 2D orthogonal temperature gradient by switching on the diode current. Moreover, we find minor difference between the experimentally probed temperature map and our finite-element simulation. We therefore replaced Fig. 2(c) in the main text with the updated temperature map in our revised manuscript, and added a section describing our temperature calibration in our updated supplementary information.

(1) First, the thermal gradient inside the MnSi sample, a key parameter in this experiment, is not experimentally determined. As an alternative way, the authors provide the simulation of the thermal gradient inside the MnSi sample as shown in Fig. 2c and discuss the detailed physics based on this simulation, but this is critical problem. We know that the simulation is useful tool if we prepare the ideal experimental setting. In short, in the case of this manuscript, if the authors discuss the thermal gradient inside the Cu holder, where two thermometers on cool and heat sides are attached, I agree the simulated thermal gradient is more reliable. However, in the current experimental setting, the authors discuss the thermal gradient inside MnSi sample attached by probably the mechanical clamping on the Cu sample holder (there is no detailed information about clamping in MS and Suppl.). In the case of the mechanical clamping, we need to consider the possibility that the thermal contact between the MnSi sample and Cu holder is not good by the thermally point contact problem. In that case, the thermal gradient inside the MnSi sample should be more complex than the ideal case. (If the authors use soldering or welding to connect the MnSi sample and Cu holder, the thermal contact is much better.)

We wish the referee find our method of the temperature measurement inside of MnSi crystal sound. In regard with the mechanical clamping, we fully agree with the referee that the thermal contact is not ideal. However, it is worth noting that the mechanical clamping only applies on the front side of the crystal, while the back side of the crystal has been well suited inside of the slot, which has good thermal contact with the holder's material. We believe this is the reason that causes the temperature difference between the front and the back side, which is the key that exerts g_{\perp} in our experiment. A good evidence can be seen by mounting the crystal upside down, it is found that the clamping side is always hotter than the other side. Therefore, we adopted the previous clamping manner to mount the sample in the recent experiment, in order to maintain the secondary g_{\perp} gradient.

(2) Secondly, I consider the possibility of the bad thermal contact between the MnSi sample and Cu holder has been already shown in the experimental data. In Fig. 3e and 3f, the intensities of the SANS for 1 and 2 of diffractions increase after switching off the heater power. There is possibility that the origin of this increase is the thermal gradient between the Cu holder and MnSi sample after switching off the heater due to the bad thermal contact with them. The authors should eliminate this possibility.

We thank for the referee's comments about the possible explanation of the increment of the peak intensity after the diode current is switched off. The referee is correct that the bad thermal contact due to the mechanical clamping could be the reason. Nevertheless, after careful examinations, we can exclude such cause with confidence.

First, at $\bar{T} = 28.05$ K, after a full cycle of the diode current ON and OFF, we found that the temperature readings from the four channels return to be as stable as 28.05 K over time. This suggests that the temperature configuration inside of the MnSi crystal recovers to be homogenous by switching off the diode current. Second, if the temperature map becomes wrongly defined inside of the sample, one would expect the decrement/increment of the peak's intensity for all six peaks, or at least the behaviour would not be reproducible. On the contrary, after the diode is switched off, we observed three of the peaks are getting stronger, while three peaks are getting weaker. Moreover, such behaviour is highly reproducible. We thus can exclude the possibly of the bad thermal contact, while we ascribe such inertia behaviour as an intrinsic effect, as captured by our numerical simulations in Fig. 5b from our main text.

Reviewers' Comments:

Reviewer #1:

Remarks to the Author:

I am very pleased to see that the authors considered all my previous comments. They added and discussed the references I suggested. Most importantly, they conducted the simulations which unambiguously confirm the theoretical results.

I believe the manuscript is now ready for publication in Nat Comm.

Minor comment: There is a typo in line 69: Weibenhofer \diamond Weißenhofer, needs a German special character.

Reviewer #2:

Remarks to the Author:

In this revised manuscript, authors have addressed all my comments and comments from the other two reviewers satisfactorily. I expect the present manuscript could stimulate more exciting discoveries on the thermal aspects of skyrmion dynamics. I suggest its publication in Nature Communications.

Reviewer #3:

Remarks to the Author:

The authors correctly improve their manuscript with following my comment. Thus, I conclude the current manuscript is suitable for the publication in Nature communications.